# Possibility of Venous Serum Cl^−^ Concentration ([Cl^−^]_s_) as a Marker for Human Metabolic Status: Correlation of [Cl^−^]_s_ to Age, Fasting Blood Sugar (FBS), and Glycated Hemoglobin (HbA1c)

**DOI:** 10.3390/ijms222011111

**Published:** 2021-10-15

**Authors:** Yoshinori Marunaka, Katsumi Yagi, Noboru Imagawa, Hironori Kobayashi, Masaru Murayama, Asami Minamibata, Yoshiaki Takanashi, Takashi Nakahari

**Affiliations:** 1Medical Research Institute, Kyoto Industrial Health Association, Kyoto 604-8472, Japan; ktsmyg@yahoo.co.jp (K.Y.); imagawa@hokenkai.jp (N.I.); kobayasi@hokenkai.jp (H.K.); murayama@hokenkai.jp (M.M.); asami-minamibata@hokenkai.jp (A.M.); yosiaki-takanasi@hokenkai.jp (Y.T.); nakahari@fc.ritsumei.ac.jp (T.N.); 2Research Organization of Science and Technology, Ritsumeikan University, Kusatsu 525-8577, Japan; 3Graduate School of Medical Science, Kyoto Prefectural University of Medicine, Kyoto 802-8566, Japan; 4Luis Pasteur Center for Medical Research, Kyoto 606-8225, Japan

**Keywords:** Cl^−^, FBS, HBA1c, pH, HCO_3_^−^, metabolism

## Abstract

The HCO_3_^−^ concentration in venous serum ([HCO_3_^−^]_s_) is a factor commonly used for detecting the body pH and metabolic conditions. To exactly detect [HCO_3_^−^]_s_, the venous CO_2_ pressure should be kept as it is in the vein. The [HCO_3_^−^]_s_ measurement is technically complicated to apply for huge numbers of almost heathy persons taking only basic medical examinations. The summation of [HCO_3_^−^]_s_ and the venous serum Cl^−^ concentration ([Cl^−^]_s_) is approximately constant; therefore, we studied if [Cl^−^]_s_ could be a marker detecting metabolic conditions instead of [HCO_3_^−^]_s_. Venous blood was obtained from persons taking basic medical examinations (the number of persons = 107,630). Older persons showed higher values of [Cl^−^]_s_, fasting blood sugar (FBS), and glycated hemoglobin (HbA1c) than younger ones. [Cl^−^]_s_ showed positive correlation to age and negative correlation to FBS and HBA1c. The negative correlation of [Cl^−^]_s_ to FBS/HbA1c was obvious in persons with high FBS/HbA1c, leading us to an idea that persons with high FBS/HbA1c show high [HCO_3_^−^]_s_, which might be caused by low activity of carbonic anhydrase in the lung observed in persons with diabetes mellitus under acidotic conditions. Taken together, an easily measured serum electrolyte, [Cl^−^]_s_, could be a useful marker estimating metabolic conditions.

## 1. Introduction

The metabolism is one of the most important functions maintaining our life activities. To exactly detect the metabolic condition, we have to measure various factors such as O_2_ consumption, CO_2_ production, pH in venous serum, HCO_3_^−^ concentration in venous serum ([HCO_3_^−^]_s_), fasting blood sugar (FBS), and glycated hemoglobin (HbA1c), etc. [1,2,3,4]. However, even though O_2_ consumption and CO_2_ production are measured, certain momentary values of O_2_ consumption and CO_2_ production are not enough to estimate the metabolic condition of whole body, but continuous measurements of O_2_ consumption and CO_2_ production are required to detect the relatively chronic metabolic condition of whole body [1,2,3,4]. On the one hand, concentrations of electrolytes such as H^+^ and HCO_3_^−^ in the venous serum show the relatively chronic status of metabolic conditions [5,6,7,8,9], although acute changes in metabolic conditions would also affect H^+^ and HCO_3_^−^ concentrations in the venous serum with a time lag dependent on the degree and the time duration of the acute metabolic changes. Even though these measurements could provide crucial information on the metabolic status, these measurements require technically complicated processes. On the other hand, to obtain information on metabolic conditions of huge numbers of persons taking only basic medical examinations not including [HCO_3_^−^]_s_ measurements, we should find out another index measurable using a technique easily adaptable to huge numbers of persons. Here, we considered that the venous serum Cl^−^ concentration ([Cl^−^]_s_) could be an index indicating metabolic conditions, since Cl^−^ is an easily measurable index and [Cl^−^]_s_ changes to the opposite direction with the change in [HCO_3_^−^]_s_. We assumed that the total amount of [Cl^−^]_s_ and [HCO_3_^−^]_s_ would be approximately constant, as described as follows. The source of HCO_3_^−^ is CO_2_ produced in metabolic cells such as myocytes, hepatocytes, and renal epithelial cells, moving into erythrocytes [6,10]. The CO_2_ in erythrocytes is converted to H^+^ and HCO_3_^−^ via a carbonic anhydrase (CA)-facilitated process (CO_2_ + H_2_O ⟶ H^+^ HCO_3_^−^) [6,10,11]. The HCO_3_^−^ is excreted into the serum by exchanging HCO_3_^−^ with serum Cl^−^, which is incorporated into erythrocytes, via a Cl^−^/HCO_3_ anion exchanger (AE)-mediated process [6,10,12,13,14,15,16,17]. The AE-mediated process leads to a decrease in [Cl^−^]_s_ associated with an increase in [HCO_3_^−^]_s_. Thus, in the present study, we tried to clarify if [Cl^−^]_s_ could be an index for metabolic conditions instead of [HCO_3_^−^]_s_, although the kidneys and lungs regulate the HCO_3_^−^ concentration leading us to consider the function of the kidneys and lungs at evaluating [Cl^−^]_s_ as an index for metabolic conditions [18].

In the present study, we indicated that: (1) Older persons show higher values of [Cl^−^]_s_, FBS, and HbA1c than younger ones; (2) [Cl^−^]_s_ changes with positive correlation to the change of age and negatively correlated to the change of FBS and HbA1c with the order of correlation intensity, age > HbA1c > FBS; (3) [Cl^−^]_s_ of persons with extremely high FBS or/and HbA1c changes more negatively correlated to FBS and HbA1c than that with normal or moderately high FBS or/and HbA1c. These observations led us to the following idea: (1) Older persons show low [HCO_3_^−^]_s_ due to low production of CO_2_; (2) persons with extremely high FBS or/and HbA1c would show high [HCO_3_^−^]_s_ due to high production of CO_2_; (3) high [HCO_3_^−^]_s_ might be also caused by slow conversion of H^+^ and HCO_3_^−^ to CO_2_ and H_2_O (H^+^ + HCO_3_^−^ ⟶ CO_2_ + H_2_O) via CA-medicated processes in the lung of persons with high leveled FBS or/and HbA1c; and (4) this might be due to low activity of CA observed in capillary endothelia of the lung in diabetes mellitus (DM) patients with high leveled FBS or/and HbA1c.

## 2. Results

### 2.1. Age-Dependent Changes in Venous Serum Cl^−^ Concentration ([Cl^−^]_s_)

We firstly studied if the [Cl^−^]_s_ would change in an age-dependent manner. To clarify this point in persons taking medical examinations (the number of persons (n) = 107,630), we categorized the age of persons taking medical examinations into six groups as shown in Table 1; the number of persons (n) in each group is also shown in Table 1. The [Cl^−^]_s_ significantly increased with the age up to 60s (Figure 1), reaching a plateau value in the persons with the ages of 60s and over 70 years old (70≦); we detected no significant difference between 60s and 70≦ (Figure 1. The minimum mean value of [Cl^−^]_s_ was observed at the age <30 (104.10 mEq/L; 95% confidence interval (CI) = 104.02–104.19 mEq/L in Figure 1). On the one hand, the maximum mean value of [Cl^−^]_s_ was observed at the age 60s (105.07 mEq/L; 95% CI = 105.04–105.10 mEq/L; Figure 1) and 70≦ (105.09 mEq/L; 95% CI = 105.03–105.14 mEq/L; Figure 1): no significant difference of the mean [Cl^−^]_s_ values was observed between these two groups, 60s and 70≦ (Figure 1). The difference between the mean [Cl^−^]_s_ values at the ages of all persons in the present study is within only 1 mEq/L; i.e., the minimum and maximum mean values of [Cl^−^]_s_ among the six groups were, respectively, 104.10 and 105.09 mEq/L (Figure 1). Nevertheless, the mean value of [Cl^−^]_s_ significantly increased in an age-dependent manner up to the 60s (Figure 1). The observation shown in Figure 1 suggests that the age-dependent change in [Cl^−^]_s_ would have some physiological meanings.

We considered a possibility that [Cl^−^]_s_ could be an index indicating metabolic conditions in medical examinations based on the following reason. CO_2_ produced in metabolic cells moves into erythrocytes, and is converted to H^+^ and HCO_3_^−^ (H^+^ + HCO_3_^−^ ⟶ CO_2_ + H_2_O) via CA-medicated processes in erythrocytes. The H^+^ produced from CO_2_ is bound to hemoglobin (Hb), while the HCO_3_^−^ produced from CO_2_ in erythrocytes is excreted to the serum in blood (the extracellular space of erythrocytes) by AE expressed on the plasma membrane of erythrocytes. The AE participates in HCO_3_^−^ excretion from erythrocytes to the extracellular space (the serum in blood) and simultaneously Cl^−^ uptake into erythrocytes from the serum of blood around metabolic cells [6,10,15,16,19,20,21,22,23,24,25]. To clarify the relationship between tissue metabolisms and [Cl^−^]_s_, we studied the age-dependent change in venous serum fasting blood sugar (FBS) and HbA1c, which have correlation to tissue metabolism, although [Cl^−^]_s_ and [HCO_3_^−^]_s_ are also affected by the respiration in the lung.

### 2.2. Age-Dependent Changes in Venous Serum Fasting Blood Sugar Concentration (FBS)

We studied if FBS would change in an age-dependent manner. FBS significantly increased in an age-dependent manner up to the age 70≦ (Figure 2) similar to that in [Cl^−^]_s_, although the age-dependent increase in [Cl^−^]_s_ reached a plateau level at the age 60s (Figure 1).

### 2.3. Age-Dependent Changes in Venous Hemoglobin A1c (HbA1c)

We further studied if HbA1c would change in an age-dependent manner. HbA1c significantly increased in an age-dependent manner up to the age 70≦ as shown in Figure 3. This age-dependent phenomenon observed in HbA1c (Figure 3) seems to be similar to that in FBS. However, the increase in HbA1c from the age 60s to 70≦ (Figure 3) seems to be larger in degree than that in FBS (Figure 2). This phenomenon would be due to the increase in post-prandial blood sugar (PBS) levels of persons with age 70≦ from 60s being larger in degree than that in FBS. This is so-called “impaired glucose tolerance” caused by deficiency in insulin secretion responding to elevation of blood sugar or/and insulin resistance occurring much more severely in 70≦ than in 60s. The “impaired glucose tolerance” influences PBS but not FBS.

### 2.4. Relationship among [Cl^−^]_s_, Age, FBS and HbA1c

Although our observations indicate that [Cl^−^]_s_, FBS, and HbA1c significantly increase in an age-dependent manner, we have no information on the relationship among [Cl^−^]_s_, FBS, and HbA1c. Therefore, we tried to clarify the relationship among [Cl^−^]_s_, age, FBS, and HbA1c using Equation (1) (see Section 4.5 in Materials and Methods). [Cl^−^]_s_ showed significantly positive correlation to age ($C_{\mathrm{AFH}}^{\mathrm{Age}}$ > 0; Table 2), but significantly negative correlation to FBS or HbA1c ($C_{\mathrm{AFH}}^{\mathrm{FBS}}$ < 0 and $C_{\mathrm{AFH}}^{\mathrm{HbA}1c}$ < 0; Table 2). However, it is unclear which factor, age, FBS, or HbA1c, most effectively influenced [Cl^−^]_s_, since age, FBS, and HbA1c had different units and these factors could not be compared to each other. To clarify this point, we normalized the values of [Cl^−^]_s_, age, FBS, and HbA1c (see Section 4.6 in Materials and Methods).

The ^N^[Cl^−^]_s_-influencing coefficient of each factor, CNAFHAge, CNAFHFBS, or CNAFHHbA1c, is significantly different from each other. ^N^age had significantly a positive effect on ^N^[Cl^−^]_s_ (CNAFHAge>0 in Table 3), while both ^N^FBS and ^N^HbA1c had significantly negative effects on ^N^[Cl^−^]_s._ Among the absolute values of coefficients, CNAFHAge, CNAFHFBS, and CNAFHHbA1c, the largest one was CNAFHAge (Table 3); i.e., ^N^age was the most effective factor on ^N^[Cl^−^]_s_, ^N^HbA1c was the next effective one on ^N^[Cl^−^]_s_, and ^N^FBS was the most non-effective one influencing ^N^[Cl^−^].

We further analyzed the correlation of [Cl^−^]_s_, FBS, or HbA1c to age using the normalized data, ^N^[Cl^−^]_s_, ^N^FBS, ^N^HbA1c, and ^N^age (see Section 4.7 in Materials and Methods). All three coefficients, CNAgeAge, FNAgeAge, and HNAgeAge, were significantly larger than 0, and each coefficient was significantly different from each other (Table 4). HbA1c was the most ^N^age-dependent factor (Table 4). FBS depended on ^N^age almost similar to HbA1c, but significantly less dependent on ^N^age than HbA1c (Table 4). [Cl^−^]_s_ least depended on ^N^age (Table 4). The value of the ^N^age-dependent coefficient for ^N^[Cl^−^]_s_ (CNAgeAge; see Table 4) was smaller than that of the ^N^FBS/^N^HbA1c-independent, ^N^age-dependent coefficient for ^N^[Cl^−^]_s_ (CNAFHAge; see Table 3), since CNAgeAge contains FBS/HbA1c-dependent factors negatively influencing [Cl^−^]_s_ (see Section 4.8 in Materials and Methods).

**Table 4 ijms-22-11111-t004:** The mean value of age-dependent coefficients,CNAgeAge, FNAgeAge, and HNAgeAge, for [Cl−]Ns, FNBS, and HNbA1c, respectively, shown in Equations (3)–(5).

Coefficient	CNAgeAge	FNAgeAge	HNAgeAge
UL of 95% CI	0.1226	0.2707	0.2870
Mean	0.1167	0.2649	0.2812
LL of 95% CI	0.1108	0.2591	0.2754

CNAgeAge, FNAgeAge, and HNAgeAge are respectively the ^N^age-dependent coefficients for ^N^[Cl^−^]_s_, ^N^FBS, or ^N^HbA1c. The upper limit (UL) and the lower limit (LL) of 95% confidence interval (CI) of the mean value of the coefficient are also shown. n = 107,630.

### 2.5. Relationship between [Cl^−^]_s_ and FBS

As shown in Table 3 and Table 4, it is suggested that [Cl^−^]_s_ is negatively correlated to FBS. Therefore, we next analyzed the relationship between [Cl^−^]_s_ and FBS by categorizing FBS into three ranges, (1) FBS < 100, (2) 100 ≦ FBS < 126, and (3) 126 mg/dL ≦ FBS using Equation (9) (see Section 4.9 in Materials and Methods). The coefficient ($C_{\mathrm{FBS}}^{\mathrm{FBS}}$) in each group of 1) FBS < 100, 2) 100 ≦ FBS < 126, or 3) 126 mg/dL ≦ FBS was significantly different from 0 (Table 5). The value of $C_{\mathrm{FBS}}^{\mathrm{FBS}}$ in the group of FBS < 100 mg/dL was significantly different from that in the group of 100 ≦ FBS < 126 or 126 mg/dL ≦ FBS (i.e., FBS ≧ 100 mg/dL), while the values of $C_{\mathrm{FBS}}^{\mathrm{FBS}}$ in the groups of 100 ≦ FBS < 126, and 126 mg/dL ≦ FBS were not significantly different (Table 5). In the group of FBS < 100 mg/dL, [Cl^−^]_s_ increased as FBS was elevated, while [Cl^−^]_s_ decreased as FBS was elevated in the group of FBS ≧ 100 mg/dL (the groups of 100 ≦ FBS < 126, and 126 mg/dL ≦ FBS).

### 2.6. Relationship between [Cl^−^]_s_ and HbA1c

As shown in Table 3 and Table 4, it is suggested that [Cl^−^]_s_ is negatively correlated to HbA1c. Therefore, we next analyzed the relationship between [Cl^−^]_s_ and HbA1c by categorizing HbA1c into three ranges, (1) HbA1c < 5.6%, (2) 5.6% ≦ HbA1c < 6.5%, and (3) 6.5% ≦ HbA1c using Equation (10) (see Section 4.10 in Materials and Methods). Each coefficient, $C_{\mathrm{HbA}1c}^{\mathrm{HbA}1c}$, in the group of (1) HbA1c < 5.6%, (2) 5.6% ≦ HbA1c < 6.5% or (3) 6.5% ≦ HbA1c is significantly smaller than 0 (Table 6): i.e., $C_{\mathrm{HbA}1c}^{\mathrm{HbA}1c}$ in each HbA1c range is negative. No significant difference of $C_{\mathrm{HbA}1c}^{\mathrm{HbA}1c}$ was observed between HbA1c < 5.6% and 5.6% ≦ HbA1c < 6.5%, while $C_{\mathrm{HbA}1c}^{\mathrm{HbA}1c}$ in the group of 6.5% ≦ HbA1c was significantly different from that in the group of HbA1c < 5.6% or 5.6% ≦ HbA1c < 6.5% (i.e., HbA1c < 6.5%; Table 6). These observations indicate that persons with HbA1c ≧ 6.5% showed [Cl^−^]_s_ decreases in a significantly large degree as HbA1c were elevated compared with those with HbA1c < 6.5% (Table 6).

### 2.7. Relationship between FBS and HbA1c

We next analyzed the relationship between FBS and HbA1c using Equation (11) (see Section 4.11 in Materials and Methods), although it is well known that FBS and HbA1c show positive correlation. We also analyzed the relationship between FBS and HbA1c using the normalized data with Equation (12) (see Section 4.11 in Materials and Methods). Both $F_{\mathrm{FBS}}^{\mathrm{HbA}1c}$ and FNFBSHbA1c are significantly larger than 0 (Table 7), suggesting that HbA1c changes had a positive correlation to the change in FBS. FNFBSHbA1c enabled us to realize the relationship between FBS and HbA1c; the change of FBS in its 78% weight would influence the change in HbA1c.

We further analyzed the relationship between FBS and HbA1c by categorizing FBS into three ranges, (1) FBS < 100 mg/dL, (2) 100 mg/dL ≦ FBS < 126 mg/dL, and (3) 126 mg/dL ≦ FBS using Equations (11) and (12). Table 8 shows the analyzed results using original and normalized data. $F_{\mathrm{FBS}}^{\mathrm{HbA}1c}$ was significantly different from each other among the group, (1) FBS < 100, (2) 100 ≦ FBS < 126 mg/dL, and (3) 126 mg/dL ≦ FBS (Table 8: Tukey–Kramer’s HSD). It is obvious that FNFBSHbA1c is significantly different from each other among the group, (1) FBS < 100 mg/dL, (2) 100 mg/dL ≦ FBS < 126 mg/dL, and (3) 126 mg/dL ≦ FBS (Table 8: Tukey–Kramer’s HSD). Interestingly, $F_{\mathrm{FBS}}^{\mathrm{HbA}1c}$ in persons with FBS ≧ 100 mg/dL is much larger than that with FBS < 100 mg/dL; a similar observation is obviously seen in FNFBSHbA1c. These observations indicate that HbA1c in persons with high FBS (FBS ≧ 100 mg/dL) would be positively correlated to FBS in a much larger degree than that with normal FBS (FBS < 100 mg/dL). In another word, HbA1c in persons with normal FBS (FBS < 100 mg/dL) show relatively little correlation to FBS compared with that with high FBS (FBS ≧ 100 mg/dL).

### 2.8. Possible Mechanims of [Cl^−^]_s_ Changes by Age, FBS, and HbA1c and Clincially Significant Meanings of [Cl^−^]_s_

We indicated possible mechanisms inducing the phenomena observed in the present study and clinical significances of [Cl^−^]_s_ suggested by the observations in the present study.

#### 2.8.1. Possible Mechanisms of [Cl^−^]_s_ Changes by Age

Figure 4A shows possible mechanisms of age-dependent changes in [Cl^−^]_s_. Figure 4a shows the case of younger persons. Younger persons have normal mitochondrial function [26,27,28,29]. Glucose is metabolized into pyruvic acid, and then CO_2_ is produced from the pyruvic acid in mitochondria with normal function. The produced CO_2_ moves into erythrocytes, and is converted into H^+^ and HCO_3_^−^ via a CA-facilitated process. The HCO_3_^−^ is exchanged with serum Cl^−^ via a Cl^−^/HCO_3_ anion exchanger (AE). These processes lead to low [Cl^−^]_s_. Figure 4b shows cases of older persons. Mitochondrial function is lower in older persons compared to younger ones [26,27,28,29]. In older persons, the amount of CO_2_ produced in mitochondria becomes low due to low mitochondrial function. Thus, the amount of H^+^ and HCO_3_^−^ produced from CO_2_ becomes low. These processes keep high [Cl^−^]_s_.

#### 2.8.2. Possible Mechanisms of [Cl^−^]_s_ Changes by FBS and HbA1c

Figure 4B shows possible mechanisms of FBS/HbA1c-dependent changes in [Cl^−^]_s_. The analytical results shown in Table 5 and Table 6 indicate that [Cl^−^]_s_ would decrease as FBS or HbA1c increases in persons with FBS (≧100 mg/dL) or HbA1c of all ranges. However, we have no information on the definite reason why [Cl^−^]_s_ would decrease as FBS or HbA1c increases in almost healthy persons except cases of FBS < 100 mg/dL. A decrease in [Cl^−^]_s_ would be due to an increase in [HCO_3_^−^]_s_ converted from CO_2_ produced in metabolic cells associated with an increase in [H^+^] converted from CO_2_ [6,10,19,30,31]. This means that [HCO_3_^−^]_s_ would increase as FBS or/and HbA1c become larger via elevation of CO_2_ production except cases of FBS < 100 mg/dL under the condition with normal mitochondrial function (Figure 4B): i.e., under the normal mitochondrial function, [Cl^−^]_s_ would decrease associated with an increase [HCO_3_^−^]_s_ when FBS and HbA1c are elevated, since the elevation of FBS and HbA1c would increase glucose metabolism resulting in large production of CO_2_ under the normal mitochondrial function with normal glucose transport function across the plasma membrane of metabolic cells (Figure 4B).

On one hand, we have observed a contrary phenomenon in persons with FBS < 100 mg/dL that [Cl^−^]_s_ would increase according to elevation of FBS (Table 5) compared with the phenomenon that [Cl^−^]_s_ would decrease according to elevation of FBS or/and HbA1c in persons with FBS ≧ 100 mg/dL and all HbA1c ranges (Table 5 and Table 6). As well known, HbA1c shows the average of blood sugar (glucose) level during one–two months [32,33,34], while FBS shows literally the blood sugar level at the fasting state [32,33,34]. If [Cl^−^]_s_ would correlate to chronic metabolic states, [Cl^−^]_s_ would show stronger correlation to HbA1c than FBS. Indeed, this point is confirmed by the analytical results shown in Table 3. Further, to confirm the relationship between FBS and HbA1c, we analyzed the relationship (Table 8). FNFBSHbA1c, a coefficient of FBS influencing HbA1c using the normalized data, is much smaller in persons with FBS < 100 mg/dL than that with FBS ≧ 100 mg/dL. This means that FBS shows much stronger correlation to the average of blood glucose sugar (glucose) levels for chronic time duration indicated as HbA1c in persons with FBS ≧ 100 mg/dL than that in FBS < 100 mg/dL (Table 8). Therefore, the phenomenon of [Cl^−^]_s_ increases according to FBS elevation in persons with FBS < 100 mg/dL unlike FBS ≧ 100 mg/dL would be due to the weak correlation of FBS to chronic blood sugar levels (HbA1c) in persons with FBS < 100 mg/dL (Table 8); i.e., FBS would not strongly reflect the average of blood glucose sugar (glucose) levels unlike HbA1c in persons with the normal FBS level (FBS < 100 mg/dL). These observations on the relationship between [Cl^−^]_s_ and HbA1c indicate the following possibilities regarding the body conditions: (1) Elevation of HbA1c associated with diminution of [Cl^−^]_s_ suggests normality of mitochondrial function with hyperphagia; (2) elevation of HbA1c associated with augmentation of [Cl^−^]_s_ suggests abnormality of mitochondrial function and disorder of glucose uptake into metabolic cells mainly due to aging-induced disorders of mitochondrial function and glucose uptake into metabolic cells (Figure 4).

#### 2.8.3. Clinically Significant Meanings of [Cl^−^]_s_ Values

Based on these observations, we recognize the clinically significant meanings of low [Cl^−^]_s_ in almost healthy persons as follows: (1) the normality of glucose uptake into metabolic cells and glucose metabolism in metabolic cells; (2) appearance of slight insulin resistance via the reduction of interstitial fluid pH dependent on high HbA1c. Thus, we suggest that reduced values of [Cl^−^]_s_ could be a clinically useful marker as recognition of glucose uptake, metabolism and slight insulin resistance in almost healthy persons combining the value of HbA1c. Clinically significant meanings of [Cl^−^]_s_ values are summarized in Table 9.

## 3. Discussion

The analytical results in the present study indicate that: (1) [Cl^−^]_s_, FBS, and HbA1c significantly increase with age; (2) [Cl^−^]_s_ shows positive correlation to age, and negative correlation to FBS and HbA1c especially in persons with high FBS (≧126 mg/dL) and HbA1c (≧6.5%); (3) the most [Cl^−^]_s_-influencing factor is age among three factors, age, FBS, and HbA1c (c.f., Figure 4A summarizes age effects on [Cl^−^]_s_, FBS, and HbA1c, and Figure 4B summarizes FBS/HbA1c effects on [Cl^−^]_s_ in persons with normal mitochondrial function).

The change in [Cl^−^]_s_ would depend on the production of CO_2_ in metabolic cells such as myocytes, hepatocytes, renal epithelial cells, etc. CO_2_ produced in metabolic cells moves into erythrocytes, then CO_2_ is converted to H^+^ and HCO_3_^−^ (CO_2_ + H_2_O ⟶ H^+^ HCO_3_^−^) in erythrocytes via a CA-facilitated process [6,10]. H^+^ produced from CO_2_ in erythrocytes bounds to Hb, while HCO_3_^−^ produced from CO_2_ in erythrocytes is excreted to the serum in blood (the extracellular space of erythrocytes) via the AE-mediated process, participating in uptake of Cl^−^ into erythrocytes from the serum in blood [6,10]. CAs expressed in erythrocytes are I and II isozymes of CAs: CAI and CAII [35]. This Cl^−^ movement into erythrocytes across the plasma membrane is well-known as “Cl^−^ shift”: (1) in erythrocytes, the Cl^−^ concentration increases associated with a decrease of HCO_3_^−^ concentration; (2) in the serum, [Cl^−^]_s_ concentration decreases associated with an increase of [HCO_3_^−^]_s_. Thus, elevation of CO_2_ production in metabolic cells would increase [HCO_3_^−^]_s_ associated with a decrease of [Cl^−^]_s_ in the serum of blood [6,7,8]. Compared with younger persons, older persons show smaller O_2_ uptake due to slower O_2_ uptake kinetics [26], limitation of oxygen delivery [27], and low rates of electron transfer and O_2_ uptake in mitochondria [28]. These reports suggest that the amount of CO_2_ production would be lower in older persons than that in younger ones, since CO_2_ is produced from O_2_ in mitochondria. Further, mitochondria dysfunction appears in an age-dependent manner [28,29]. Mitochondria dysfunction leads to low O_2_ consumption resulting in low production of CO_2_. Based on these reports, the elevated [Cl^−^]_s_ observed in older persons would be due to mitochondrial dysfunction, which is also observed in persons with cancers and diabetes [36,37,38], thus [Cl^−^]_s_ continuously (even once or twice a year) measured with easy techniques would be useful as a marker detecting mitochondrial function.

Both FBS and HbA1c show the age-dependent increases (Figure 2 and Figure 3). However, the age-dependent increase of HbA1c (Figure 3) from 60s to 70≦ looks larger in degree than that of FBS (Figure 2). The larger age-dependent increase in HBA1c than FBS from 60s to 70≦ would be due to a larger increase in PBS than FBS occurring in 70≦ compared with 60≦ caused by impaired glucose tolerance or/and insulin resistance, affecting PBS but not FBS. Mitochondrial dysfunction appearing in an age-dependent manner [28,29] induces the glycolysis-based metabolic condition associated with production of large amounts of protons (H^+^), causing acidification of the interstitial fluid [6,7,8,10,19,36,37,38,39,40,41]. This acidification causes insulin resistance via reduction of insulin affinity to its receptor [6,7,8,10,19,39,40,41], resulting in a larger increase in HbA1c due to elevation of PBS compared with elevation of insulin-independently controlled FBS from the age of 60s to 70≦. The absolute value of coefficient of ^N^HbA1c influencing [Cl^−^]_s_ being a little bit but significantly larger than that of ^N^FBS (Table 3) would be explained by the characteristics of HbA1c reflecting the average blood sugar level during one–two months before the blood-sampled time [33,34,42,43] unlike FBS literally showing the fasting blood sugar level at the blood-sampled [33,34]. Therefore, based on the observation that [Cl^−^]_s_ shows a relatively stronger correlation to HbA1c than FBS, it is suggested that [Cl^−^]_s_ would depend on the average blood sugar level reflecting the metabolic condition.

In addition, we should consider cases of diabetic ketoacidosis [10,30,44]. Diabetic ketoacidosis occurs under conditions that glucose is not available as energy sources [10,30,44]. When glucose in not available as energy source, another energy source is required: e.g., a free fatty acid is one of major energy sources at unavailability of glucose. Metabolism of free fatty acids produces ketone bodies [45]. Beta-hydroxybutyric acid (CH_3_-CH(OH)-CH_2_-COOH), one of the most major ketone bodies (~70% of total ketone bodies), is produced from free fatty acids released from adipocytes [45], and then is dissociated into beta-hydroxybutyrate^−^ (CH_3_-CH(OH)-CH_2_-COO^−^) and H^+^ (CH_3_-CH(OH)-CH_2_-COOH ⟶ CH_3_-CH(OH)-CH_2_-COO^−^ + H^+^) [46]. Under this condition, little amounts of HCO_3_^−^ are produced from glucose metabolism associated with a large amount of ketone bodies such as beta-hydroxybutyrate^−^ (CH_3_-CH(OH)-CH_2_-COO^−^), the concentration of which increases in the serum. In this case, the serum HCO_3_^−^ or Cl^−^ doesn’t change unlike the case of glucose metabolism that CO_2_ produced in metabolic cells moves into erythrocytes, dissociating into HCO_3_^−^ + H^+^ via a CA-mediated process, which leads to an increased [HCO_3_^−^]_s_ and a decreased [Cl^−^]_s_ via an AE-mediated exchange pathway. The metabolism of free fatty acids produces a large amount of H^+^ dissociated from ketone bodies at glucose unavailable states, leading to acidosis; it is called normochloremic ketoacidosis with high anion gap, which occurs in patients with severe DM [10,30,44,47,48].

In addition to this explanation on the relationship among [Cl^−^]_s_, FBS and HbA1c, we should also consider another cause for an increase of [HCO_3_^−^]_s_ with elevation of FBS or/and HbA1c: i.e., the CO_2_ excretion capacity into the atmosphere through expiration should be considered [10]. Most parts of CO_2_ produced in metabolic cells are excreted into the atmosphere through expiration in the lung [10]. The decrease in amounts of CO_2_ excretion into the atmosphere causes an increase in [HCO_3_^−^]_s_. Therefore, we should consider a possibility that the amount of CO_2_ excretion would decrease as FBS or/and HbA1c are elevated. The CO_2_ produced in metabolic cells moves into erythrocytes [6,10,19,30,31]. Then, CAs facilitate the converting process of CO_2_ to H^+^ and HCO_3_^−^ (CO_2_ + H_2_O ⟶ H^+^ HCO_3_^−^) in erythrocytes: I and II isozymes of CA (CAI and CAII) are expressed in erythrocytes [35]. The HCO_3_^−^ is excreted from erythrocytes to the serum in blood (the extracellular space of erythrocytes) via the AE-mediated pathway, while the produced H^+^ bounds to Hb (c.f., Figure 4) [6,10,19,30,31]. In the lung, the Hb-bound H^+^ and HCO_3_^−^ transported into erythrocytes from the serum via the AE-mediated reversed pathway (c.f., Figure 4) are converted to CO_2_ and H_2_O (H^+^ + HCO_3_^−^ ⟶ CO_2_ + H_2_O) via a CA-facilitated pathway in erythrocytes [6,10,19,30,31]. CA is also expressed in capillary endothelia of the lung [49,50]. The CA expressed in capillary endothelia of the lung contributes to the converting process of H^+^ and HCO_3_^−^ dissolved in the serum (several percent of total produced CO_2_) to CO_2_ + H_2_O (H^+^ + HCO_3_^−^ ⟶ CO_2_ + H_2_O). The activity of CA expressed in capillary endothelia of the lung has been reported to be lower in DM patients than healthy persons [51]. These reports [49,50,51] lead us to an idea that high [HCO_3_^−^]_s_ might be also caused by slow conversion of H^+^ and HCO_3_^−^ to CO_2_ and H_2_O (H^+^ + HCO_3_^−^ ⟶ CO_2_ + H_2_O) via CA-medicated processes in the lung of persons with high leveled FBS or/and HbA1c; (2) the decelerated converting process of H^+^ and HCO_3_^−^ to CO_2_ and H_2_O in the lung would keep high [HCO_3_^−^]_s_ in DM patients; (3) the high [HCO_3_^−^]_s_ in DM patients would keep low [Cl^−^]_s_; (4) the activity of CA would become lower as FBS or/and HbA1c increase; (5) if so, [Cl^−^]_s_ would decrease associated with elevation of FBS or/and HbA1c due to the low activity of CA under the FBS/HbA1c-elevated condition; (6) the lower activity of CA in persons with high FBS or/and HbA1c might cause acidotic conditions in blood and interstitial fluids, causing the insulin resistance [6,7,8,9,10,19,30,41].

Here, we should also consider the aging effect on gas exchange in the lung [52,53,54] including disorders of gas exchange such as chronic obstructive pulmonary disease (COPD) [54]. Symptoms of COPD are well known to progress with age [52]. Patients with COPD show difficulty to excrete CO_2_ into the atmosphere [55]. At the early stage of COPD, CO_2_ retention in the body occurs due to difficulty of CO_2_ excretion into the atmosphere in the lung [55]. Disorders of gas exchange cause low O_2_ availability in metabolic cells associated with low CO_2_ production, resulting in reduction of life activity due to low energy (ATP) supply [52,53,54,55]. Patients suffering from severe COPD would show dyspnea, therefore it is relatively easy to diagnose COPD using various diagnostic devices such as CT scan, etc. [55]. However, it is difficult to diagnose COPD or find symptoms of COPD especially at the early stage. Therefore, [Cl^−^]_s_ could be a screening maker to find out patients staying in a very early stage of COPD just by taking basic medical examinations adaptable for huge numbers of persons, although confirmed diagnosis for COPD definitely requires advanced medical diagnostic devices such as CT scan.

In addition to aging effects on the lung function, we should also consider aging effects on the kidney function. Aging decreases glomerular filtration rate (GFR) [56,57]. The age-dependent decrease in GFR diminishes the filtrating amount of serum Na^+^ and Cl^−^ [56,57], stimulating the secretion of renin followed by activation of the renin-angiotensin-aldosterone (RAA) system [57]. Thus, the activation of RAA system caused by the age-dependent decrease in GFR would be considered as another cause of [Cl^−^]_s_ increases with age.

In the present study, we analyzed the correlation among [Cl^−^]_s_, age, FBS, and HbA1c, and tried to clarify the physiological and/or pathophysiological meanings of the change in [Cl^−^]_s_. We especially focused on the relationship between the metabolic condition and [HCO_3_^−^]_s_ by measuring [Cl^−^]_s_. To clarify this point, [HCO_3_^−^]_s_ should be ideally measured. However, most persons showing no or little health problems without any serious symptoms usually take only basic medical examinations without [HCO_3_^−^]_s_ measurements due to its technical complication. Therefore, under the condition, an easily measurable index, [Cl^−^]_s_, would be useful to estimate [HCO_3_^−^]_s_ reflecting metabolic conditions adaptable to huge numbers of persons.

## 4. Materials and Methods

### 4.1. Subjects

Data were obtained from persons taking medical examinations at Kyoto Industrial Health Association from 1 April 2011 to 31 March 2017. Written information regarding the present study was provided on WEB of Kyoto Industrial Health Association announcing to persons taking medical examination that they can opt out their own data from the present study. The number (n) of the persons participating in the present study was 107,630; the average of age, 51.61 ± 0.04 (mean ± standard error) years old (18–96); male, n = 71,423, 51.76 ± 0.04 (mean ± standard error) years old (18–96); female, n = 36,207, 51.26 ± 0.06 (mean ± standard error) years old (18–89).

### 4.2. Fasting Blood Samples

Blood samples were obtained from veins of persons with fasting for more than 5 h who took medical examinations at Kyoto Industrial Health Association. We excluded persons taking any DM treatments.

### 4.3. Measurements of [Cl^−^]_s_, FBS and HbA1c

[Cl^−^]_s_, FBS, and HbA1c were measured at the laboratory of Kyoto Industrial Health Association. [Cl^−^]_s_ was measured using a Cl^−^-selective electrode, A&T Corporation, Yokohama 221-0056, Japan. HbA1c was assayed using high-performance liquid chromatography and was expressed as a National Glycohemoglobin Standardization Program unit.

### 4.4. Statistical Analysis

The statistical analysis was performed by a software, JMP 8.0 using Tukey–Kramer’s honestly significant difference (HSD). Data are shown as the mean values with the upper and lower limits of the 95% confidence interval (CI) of the mean values except the presentation of age.

### 4.5. Relationship among [Cl^−^]_s_, Age, FBS and HbA1c

The relationship among [Cl^−^]_s_, age, FBS, and HbA1c was analyzed assuming that the following equation would hold.
(1)$${\left[{\mathrm{Cl}}^{-}\right]}_{s}=C_{\mathrm{AFH}}^{\mathrm{Age}}\;\mathrm{age}+C_{\mathrm{AFH}}^{\mathrm{FBS}}\;\mathrm{FBS}+C_{\mathrm{AFH}}^{\mathrm{HbA}1c}\;\mathrm{HbA}1c+C_{\mathrm{AFH}}^{\mathrm{Int}}$$

Here, $C_{\mathrm{AFH}}^{\mathrm{Age}}$, $C_{\mathrm{AFH}}^{\mathrm{FBS}}$, and $C_{\mathrm{AFH}}^{\mathrm{HbA}1c}$ are respectively [Cl^−^]_s_-influencing coefficients of age, FBS, and HbA1c; $C_{\mathrm{AFH}}^{\mathrm{Int}}$ is the intersection value of [Cl^−^]_s_ at age, FBS, and HbA1c = 0; $C_{\mathrm{AFH}}^{\mathrm{Age}}$, $C_{\mathrm{AFH}}^{\mathrm{FBS}}$, $C_{\mathrm{AFH}}^{\mathrm{HbA}1c}$, and $C_{\mathrm{AFH}}^{\mathrm{Int}}$ are constant.

### 4.6. The Relationship among Normalized Data, ^N^[Cl^−^]_s_, ^N^Age, ^N^FBS, and ^N^HbA1c

Age, FBS, and HbA1c had different units; therefore, it was impossible to determine which factor, age, FBS, or HbA1c, most effectively influences [Cl^−^]_s_. To clarify this point, we normalized the values of [Cl^−^]_s_, age, FBS, and HbA1c by setting each mean value of [Cl^−^]_s_, age, FBS or HbA1c = 0 with each standard deviation = 1. Here, we respectively represent the normalized data of [Cl^−^]_s_, age, FBS, and HbA1c as ^N^[Cl^−^]_s_, ^N^age, ^N^FBS, and ^N^HbA1c. The relationship among [Cl^−^]_s_, age, FBS, and HbA1c was analyzed assuming that the following equation would hold.
(2)[Cl−]Ns=CNAFHAge aNge+CNAFHFBS FNBS+CNAFHHbA1c HNbA1c+CNAFHInt

Here, CNAFHAge, CNAFHFBS, and CNAFHHbA1c are respectively ^N^[Cl^−^]_s_-influencing coefficients of ^N^age, ^N^FBS, and ^N^HbA1c, and INAFHInt is the intersection value of ^N^[Cl^−^] at ^N^age, ^N^FBS, and ^N^HbA1c = 0; CNAFHAge, CNAFHFBS, CNAFHHbA1c, and CNAFHInt are constant.

### 4.7. Correlation of [Cl^−^]_s_, FBS or HbA1c to Age Using the Normalized Data, ^N^[Cl^−^]_s_, ^N^FBS, ^N^HbA1c, and ^N^Age

The analysis was performed using Equations (3)–(5), respectively.
(3)[Cl−]Ns=CNAgeAge aNge+CNAgeInt=CNAgeAge aNge
(4)[FBS]N=FNAgeAge aNge+FNAgeInt=FNAgeAge aNge
(5)[HbA1c]N=HNAgeAge aNge+HNAgeInt=HNAgeAge aNge

Here, CNAgeAge, FNAgeAge, and HNAgeAge are respectively the ^N^age-dependent coefficients for ^N^[Cl^−^]_s_, ^N^FBS, or ^N^HbA1c, and CNAgeInt, FNAgeInt, and HNAgeInt are respectively the intersection values of ^N^[Cl^−^]_s_, ^N^FBS, and ^N^HbA1c at ^N^age = 0: CNAgeInt, FNAgeInt or HNAgeInt (the intersection value of ^N^[Cl^−^]_s_, ^N^FBS, or ^N^HbA1c at ^N^age = 0) would be ideally 0, since all the mean values of [Cl^−^]_s_, age, FBS, and HbA1c were normalized to be = 0.

### 4.8. Age-Dependent Factor and FBS/HbA1c-Dependent Factor Influencing ^N^[Cl^−^]_s_

Substituting Equations (3)–(5) into Equation (2), the following equation was obtained (CNAFHInt = 0).
(6)[Cl−]Ns=CNAFHAge aNge+CNAFHFBS FNBS+CNAFHHbA1c HNbA1c+CNAFHInt=CNAFHAge aNge+CNAFHFBS FNAgeAge aNge+CNAFHHbA1c HNAgeAge aNge=(CNAFHAge+CNAFHFBS FNAgeAge+CNAFHHbA1c HNAgeAge) aNge=CNAgeAge aNge 

Thus, Equation (7) functions.
(7)CNAgeAge=CNAFHAge+CNAFHFBS FNAgeAge+CNAFHHbA1c HNAgeAge

This means that the ^N^age-dependent coefficient (CNAgeAge) for [Cl^−^]_s_ consists of CNAFHAge, CNAFHFBS FNAgeAge and CNAFHHbA1c HNAgeAge. Here, CNAFHAge is the ^N^FBS/^N^HbA1c-independent, ^N^age-dependent coefficient for ^N^[Cl^−^]_s_; CNAFHFBS FNAgeAge, the ^N^FBS/^N^age-dependent coefficient for ^N^[Cl^−^]_s_; CNAFHHbA1c HNAgeAge, the ^N^HbA1c/^N^age-dependent coefficient for ^N^[Cl^−^]_s_. Equation (8) functions, since CNAgeAge>0, CNAFHAge>0, CNAFHFBS<0 FNAgeAge>0, CNAFHHbA1c<0, and HNAgeAge>0 in Equation (7) (see Table 3 and Table 4).
(8)CNAgeAge<CNAFHAge

### 4.9. The Relationship between [Cl^−^]_s_ and FBS

The relationship between [Cl^−^]_s_ and FBS was analyzed using Equation (9).
(9)$${\left[{\mathrm{Cl}}^{-}\right]}_{s}=C_{\mathrm{FBS}}^{\mathrm{FBS}}\;\mathrm{FBS}+I_{\mathrm{FBS}}^{\mathrm{FBS}}$$

Here, $C_{\mathrm{FBS}}^{\mathrm{FBS}}$ is a [Cl^−^]_s_-influencing coefficient of FBS, $I_{\mathrm{FBS}}^{\mathrm{FBS}}$ is the intersection value of [Cl^−^]_s_ at FBS = 0, and $C_{\mathrm{FBS}}^{\mathrm{FBS}}$ and $I_{\mathrm{FBS}}^{\mathrm{FBS}}$ are constant.

### 4.10. The Relationship between [Cl^−^]_s_ and HbA1c

The relationship between [Cl^−^]_s_ and HbA1c was analyzed using Equation (10).
(10)$${\left[{\mathrm{Cl}}^{-}\right]}_{s}=C_{\mathrm{HbA}1c}^{\mathrm{HbA}1c}\;\mathrm{HbA}1c+I_{\mathrm{HbA}1c}^{\mathrm{HbA}1c}$$

Here, $C_{\mathrm{HbA}1c}^{\mathrm{HbA}1c}$ is a coefficient of HbA1c influencing [Cl^−^]_s_, $I_{\mathrm{HbA}1c}^{\mathrm{HbA}1c}$ is the intersection value of [Cl^−^]_s_ at HbA1c = 0, and $C_{\mathrm{HbA}1c}^{\mathrm{HbA}1c}$ and $I_{\mathrm{HbA}1c}^{\mathrm{HbA}1c}$ are constant.

### 4.11. Relationship between FBS and HbA1c

The relationship between FBS and HbA1c was analyzed using Equation (11), and also using the normalized data with Equation (12).
(11)$$\mathrm{HbA}1c=F_{\mathrm{FBS}}^{\mathrm{HbA}1c}\;\mathrm{FBS}+I_{\mathrm{FBS}}^{\mathrm{HbA}1c}$$
(12)HNbA1c=FNFBSHbA1c FNBS+INFBSHbA1c

Here, $F_{\mathrm{FBS}}^{\mathrm{HbA}1c}$ is a HbA1c-influencing coefficient of FBS, $I_{\mathrm{FBS}}^{\mathrm{HbA}1c}$ is the intersection value of HbA1c at FBS = 0, FNFBSHbA1c is HNbA1c-influencing coefficient of FNBS, $I_{\mathrm{FBS}}^{\mathrm{HbA}1c}$ is the intersection value of HNbA1c at FNBS = 0, and $F_{\mathrm{FBS}}^{\mathrm{HbA}1c}$, $I_{\mathrm{FBS}}^{\mathrm{HbA}1c}$, FNFBSHbA1c, and INFBSHbA1c are constant.

## 5. Conclusions

The present study indicates that: (1) the values of [Cl^−^]_s_, FBS, and HbA1c are larger in older persons than younger ones; (2) [Cl^−^]_s_ shows positive correlation to age, and negative correlation to FBS and HbA1c especially in persons with high FBS (≧126 mg/dL) and HbA1c (≧6.5%); (3) the most [Cl^−^]_s_-influencing factor is “age” among three factors, age, FBS, and HbA1c. [Cl^−^]_s_ would be a marker of metabolism and insulin resistance, and show mitochondrial function combining information on FBS/HbA1c. Figure 4 and Table 9 summarize the conclusion obtained from the present study.

## Figures and Tables

**Figure 1 ijms-22-11111-f001:**
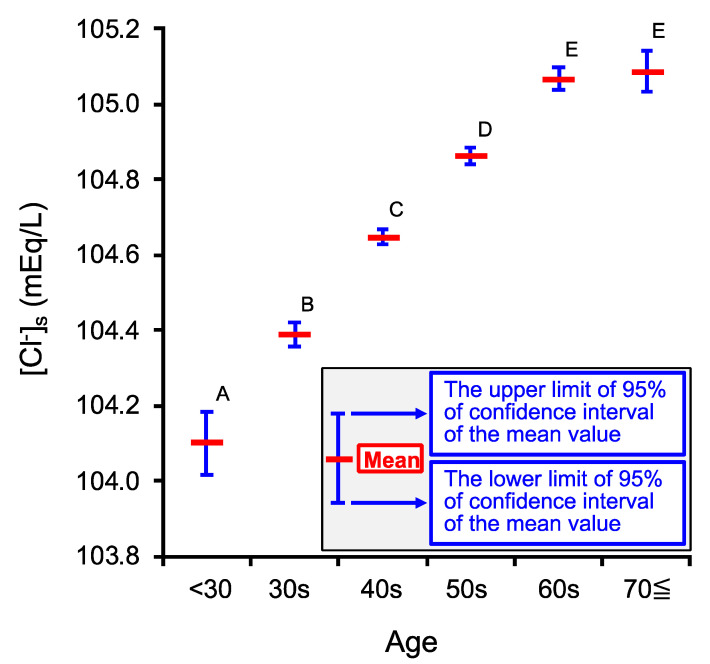
Age-dependent changes in venous serum Cl^−^ concentration ([Cl^−^]_s_). The ages of persons taking medical examinations were categorized into six groups as shown in Table 1. Red horizontal bars at the persons’ ages show the mean [Cl^−^]_s_ values of persons at the ages. The upper and lower blue horizontal bars at the persons’ ages show respectively the upper and lower limits of the 95% confidence interval (CI) for the mean [Cl^−^]_s_ values of persons at the ages. [Cl^−^]_s_ increased in an age-dependent manner up to the 60s. Labels A, B, C, D, and E show the statistical difference: the mean [Cl^−^]_s_ values of the groups labeled with different characters are significantly different from each other at a level of *p* < 0.05, while the mean [Cl^−^]_s_ values of the groups labeled with the same character are not significantly different at a level of *p* ≧ 0.05 (the mean [Cl^−^]_s_ values of persons’ age = 60s and 70≦ were not significantly different). The statistical test was performed by Tukey–Kramer’s honestly significant difference (HSD).

**Figure 2 ijms-22-11111-f002:**
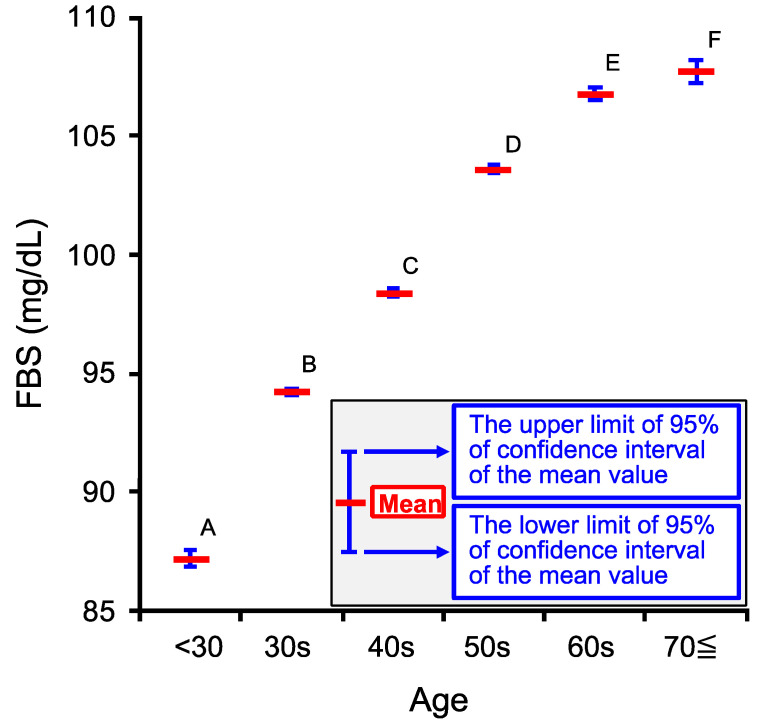
Age-dependent changes in venous serum fasting blood sugar concentration (FBS). The ages of persons taking medical examinations were categorized into six groups as shown in Table 1. Red horizontal bars show the mean values of FBS of persons at the ages. The upper and lower blue horizontal bars at the persons’ ages show respectively the upper and lower limits of the 95% confidence interval (CI) for the mean values of FBS of persons at the ages. FBS increased in an age-dependent manner up to 70 years old (70≦). Labels A, B, C, D, E and F show the statistical difference: the mean values of FBS of the groups labeled with different characters are significantly different from each other at a level of *p* < 0.05. The statistical test was performed by Tukey–Kramer’s HSD.

**Figure 3 ijms-22-11111-f003:**
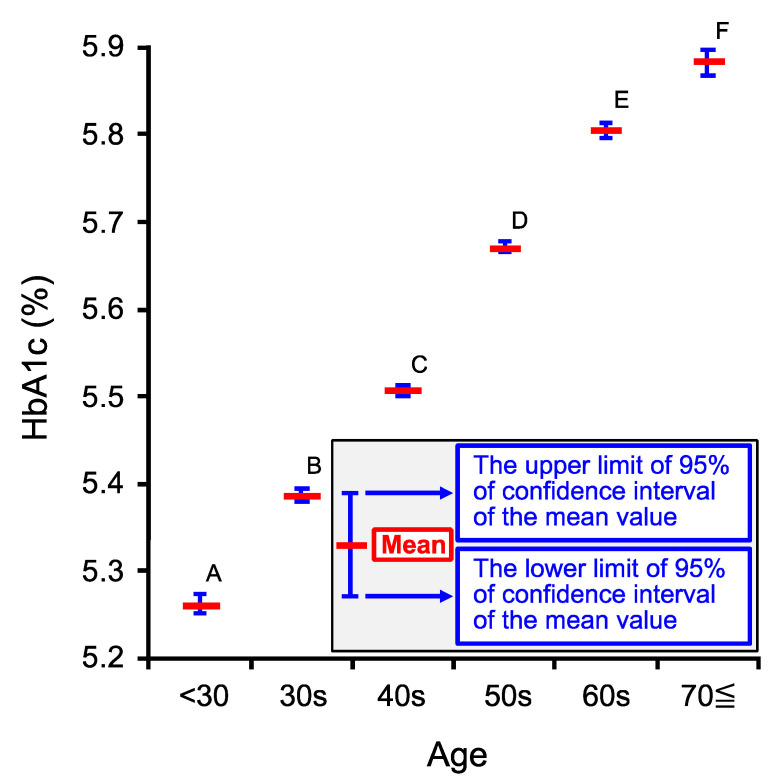
Age-dependent changes in venous HbA1c. The ages of persons taking medical examinations were categorized into six groups as shown in Table 1. Red horizontal bars at the persons’ ages show the mean values of HbA1c of persons at the ages. The upper and lower blue horizontal bars at the persons’ ages show respectively the upper and lower limits of the 95% confidence interval (CI) for the mean values of HbA1c of persons at the ages. The HbA1c increased in an age-dependent manner up to 70 years old (70≦). Labels, A, B, C, D, E, and F show the statistical difference: The mean values of HbA1c of the groups labeled with different characters are significantly different from each other at a level of *p* < 0.05. The statistical test was performed by Tukey–Kramer’s HSD.

**Figure 4 ijms-22-11111-f004:**
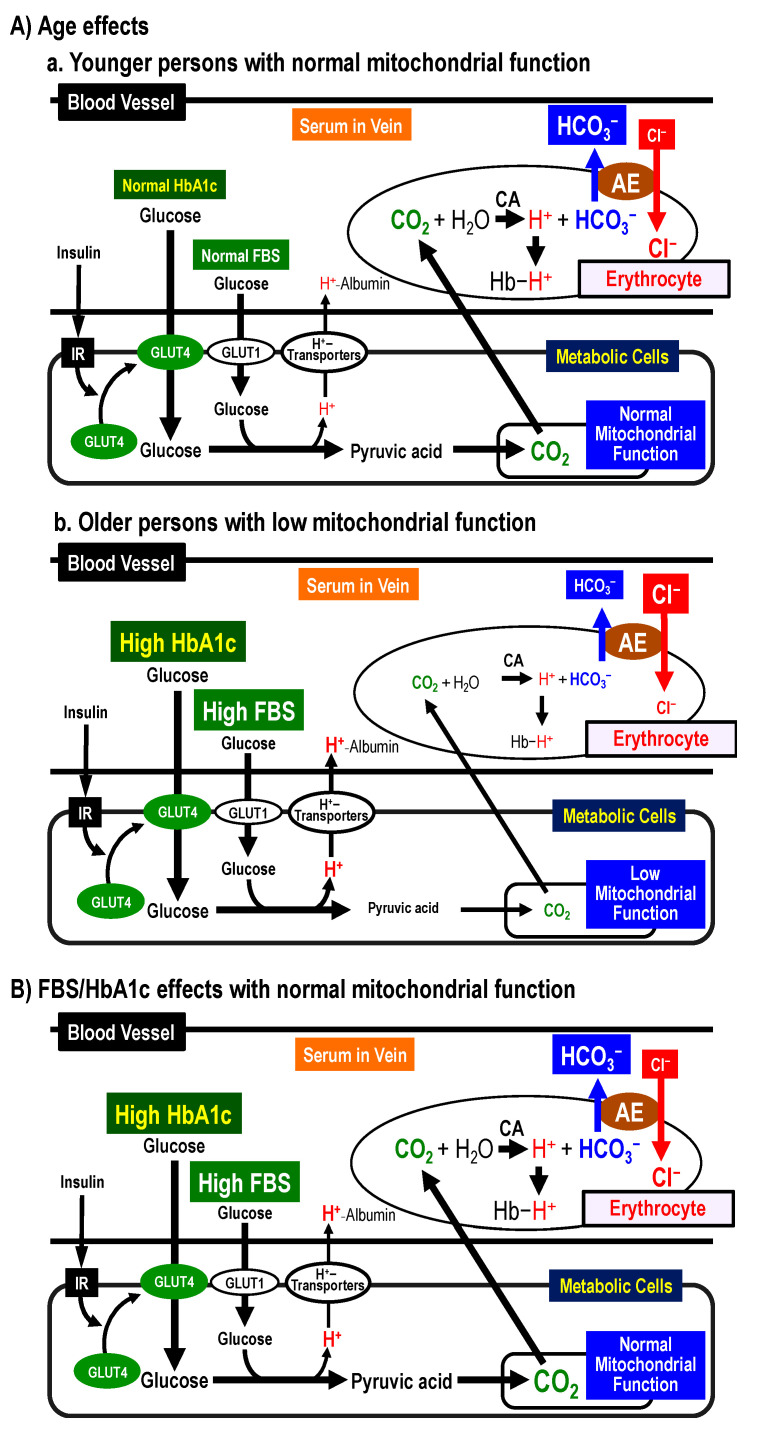
Summary. (**A**) Age effects on [Cl^−^]_s_. (**a**) Younger persons with normal mitochondrial function. Glucose is metabolized into pyruvic acid, and then CO_2_ is produced from the pyruvic acid in mitochondria with normal function. The produced CO_2_ moves into erythrocytes, and is converted into H^+^ and HCO_3_^−^ via a CA-facilitated process. The HCO_3_^−^ is exchanged with serum Cl^−^ via a Cl^−^/HCO_3_ anion exchanger (AE). These processes lead to low [Cl^−^]_s_. (**b**) Older persons with low mitochondrial function. The amount of CO_2_ produced in mitochondria becomes low due to low mitochondrial function. Thus, the amount of H^+^ and HCO_3_^−^ produced from CO_2_ becomes low. These processes keep high [Cl^−^]_s_. (**B**) FBS/HbA1c effects on [Cl^−^]_s_. with normal mitochondrial function. Glucose is metabolized into pyruvic acid, and then CO_2_ is produced from the pyruvic acid in mitochondria with normal function. The produced CO_2_ moves into erythrocytes, and is converted into H^+^ and HCO_3_^−^ via a CA-facilitated process. The HCO_3_^−^ is exchanged with serum Cl^−^ via a Cl^−^/HCO_3_ anion exchanger (AE). In cases of high FBS/HBA1c with normal mitochondrial function, large amounts of CO_2_ are produced, resulting in production of large amounts of HCO_3_^−^. These processes lead to low [Cl^−^]_s_.

**Table 1 ijms-22-11111-t001:** Age category and the number of persons in each category.

Age Category	<30	30s	40s	50s	60s	70≦
Meaning of Age Category (years old)	Age < 30	30 ≦ Age < 40	40 ≦ Age < 50	50 ≦ Age < 60	60 ≦ Age < 70	70 ≦ Age
Number of persons (n)	1878	14,300	35,457	29,175	20,344	6476

We categorized the age of persons taking medical examinations into six groups; (1) younger than 30 years old (<30), (2) equal to or older than 30 years old and younger than 40 years old (30s), (3) equal to or older than 40 years old and younger than 50 years old (40s), (4) equal to or older than 50 years old and younger than 60 years old (50s), (5) equal to or older than 60 years old and younger than 70 years old (shown as 60s), and (6) equal to or older than 70 years old (shown as 70≦).

**Table 2 ijms-22-11111-t002:** The mean values of coefficients in Equation (1) for the relationship among [Cl^−^]_s_, age, FBS, and HbA1c.

Coefficient	$C_{AFH}^{Age}$ (mEq/L/year)	$C_{AFH}^{FBS}$ (mEq/L/>mg/dL)	$C_{AFH}^{HbA1c}$ (mEq/L/%)	$C_{AFH}^{Int}$ (mEq/L)
UL of 95% CI	0.0312	−0.00727	−0.311	106.1
Mean	0.0300	−0.00837	−0.345	106.0
LL of 95% CI	0.0289	−0.00947	−0.379	105.9

$C_{\mathrm{AFH}}^{\mathrm{Age}}$, $C_{\mathrm{AFH}}^{\mathrm{FBS}}$, and $C_{\mathrm{AFH}}^{\mathrm{HbA}1c}$ are respectively [Cl^−^]_s_-influencing coefficients of age, FBS, and HbA1c; $C_{\mathrm{AFH}}^{\mathrm{Int}}$ is the intersection value of [Cl^−^]_s_ at age, FBS, and HbA1c = 0. The upper limit (UL) and the lower limit (LL) of 95% confidence interval (CI) of the mean value of the coefficient are also shown. n = 107,630.

**Table 3 ijms-22-11111-t003:** The mean value of the coefficient in Equation (2) for the relationship among [Cl^−^]_s_, age, FBS, and HbA1c using normalized data of [Cl^−^]_s_, age, FBS, and HbA1.

Coefficient	CNAFHAge (mEq/L/year)	CNAFHFBS (mEq/L/mg/dL)	CNAFHHbA1c (mEq/L/%)	CNAFHInt (mEq/L)
UL of 95% CI	0.1693	−0.0626	−0.0875	0.0059
Mean	0.1631	−0.0721	−0.0970	0.0000
LL of 95% CI	0.1569	−0.0816	−0.1065	−0.0059

CNAFHAge, CNAFHFBS, and CNAFHHbA1c are respectively ^N^[Cl^−^]_s_-influencing coefficients of ^N^age, ^N^FBS, and ^N^HbA1c, and INAFHInt is the intersection value of ^N^[Cl^−^] at ^N^age, ^N^FBS, and ^N^HbA1c = 0. The upper limit (UL) and the lower limit (LL) of the 95% confidence interval (CI) of the mean value of the coefficient are also shown. n = 107,630.

**Table 5 ijms-22-11111-t005:** The mean value of the coefficient in Equation (9) for the relationship between [Cl^−^]_s_ and FBS.

FBS (mg/dL)		FBS < 100	100 ≦ FBS < 126	126 ≦ FBS
n		59,922	41,633	6075
$C_{\mathrm{FBS}}^{\mathrm{FBS}}$	UL of 95% CI	0.0281	−0.0128	−0.0186
	Mean	0.0252	−0.0161	−0.0202
	LL of 95% CI	0.0222	−0.0194	−0.0219

$C_{\mathrm{FBS}}^{\mathrm{FBS}}$ is a [Cl^−^]_s_-influencing coefficient of FBS. The upper limit (UL) and the lower limit (UL) of 95% confidence interval (CI) of the mean value of the [Cl^−^]_s_-influencing coefficient of FBS in persons whose FBS was categorized into each range are also shown. Total number = 107,630.

**Table 6 ijms-22-11111-t006:** The mean value of the coefficient in Equation (9) for the relationship between [Cl^−^]_s_ and HbA1c.

HbA1c (%)		HbA1c < 5.6	5.6 ≦ HbA1c < 6.5	6.5 ≦ HbA1c
n		57,189	44,699	5742
$C_{\mathrm{HbA}1c}^{\mathrm{HbA}1c}$	UL of 95% CI	−0.0188	−0.1394	−0.5794
	Mean	−0.1082	−0.2332	−0.6312
	LL of 95% CI	−0.1976	−0.3269	−0.6830

$C_{\mathrm{HbA}1c}^{\mathrm{HbA}1c}$ is a coefficient of HbA1c influencing [Cl^−^]_s_. The upper limit (UL) and the lower limit (LL) of 95% confidence interval (CI) of the mean value of the coefficient of FBS influencing [Cl^−^]_s_ in persons whose [FBS] was categorized into each range are shown.

**Table 7 ijms-22-11111-t007:** The mean values of coefficients in Equations (11) and (12) for the relationship between FBS and HbA1c.

Coefficient	$F_{FBS}^{HbA1c}$	FNFBSHbA1c
UL of 95% CI	0.021725	0.787159
Mean	0.021462	0.783447
LL of 95% CI	0.021199	0.779735

$F_{\mathrm{FBS}}^{\mathrm{HbA}1c}$ is a HbA1c-influencing coefficient of FBS; FNFBSHbA1c is HNbA1c-influencing coefficient of FNBS. The upper limit (UL) and the lower limit (LL) of 95% confidence interval (CI) of the mean value of the coefficient are also shown. n = 107,630.

**Table 8 ijms-22-11111-t008:** The mean value of the coefficient in Equations (11) and (12) for the relationship between [Cl^−^]_s_ and FBS.

FBS (mg/dL)		FBS < 100	100 ≦ FBS < 126	126 ≦ FBS
n		59,922	41,633	6075
$F_{\mathrm{FBS}}^{\mathrm{HbA}1c}$	UL of 95% CI	0.0114	0.0266	0.0278
	Mean	0.0110	0.0261	0.0273
	LL of 95% CI	0.0106	0.0256	0.0267
FNFBSHbA1c	UL of 95% CI	0.3500	0.8146	0.8518
	Mean	0.3374	0.7985	0.8351
	LL of 95% CI	0.3247	0.7824	0.8185

The upper limit (UL) and the lower limit (UL) of 95% confidence interval (CI) of the mean value of the [Cl^−^]_s_-influencing coefficient of FBS in persons whose FBS was categorized into each range are also shown.

**Table 9 ijms-22-11111-t009:** Clinically significant meanings of [Cl^−^]_s_ values HbA1c.

	Low [Cl^−^]_s_		High [Cl^−^]_s_	
	Normal HbA1c	High HbA1c	Normal HbA1c *	High HbA1c
Glucose metabolism (Mitochondrial function)	Normal	Normal	Low	Low
Insulin resistance	−	+	+	++

Insulin resistance: −, no insulin resistance; +, slight insulin resistance; ++, a little bit severe insulin resistance. * High HbA1c with high [Cl^−^]_s_, this status would be caused by diet with low carbohydrates.

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
