# Peer review of "Possibility of Venous Serum Cl− Concentration ([Cl−]s) as a Marker for Human Metabolic Status: Correlation of [Cl−]s to Age, Fasting Blood Sugar (FBS), and Glycated Hemoglobin (HbA1c)"

_ijms, 2021, doi:10.3390/ijms222011111_

Round 1
Reviewer 1 Report
The manuscript deals with serum [Cl] to be established as a marker for human metabolic status, rather than the imprecise analyses based on pH [or [HCO3]. Serum [Cl] was found to be a ‘remarkably predictable [and reliable] outcomes for prediction of metabolic status.’ This reviewer would like to point out the following comments for the sake of improving the manuscript for the readers to easily grasp/understand the authors’ findings/conclusions. It is hoped by this reviewer that the authors try to clarify and improve the readability and fortify their conclusions, but it is left for the editor and the authors to decide/follow what this reviewer is suggesting as follows:
- It would be essential to describe as to why the authors chose an independent variable as the age group, rather than age itself. One way to approach this suggestion might be to show the raw data using the age itself as the independent variable, as this reviewer thinks that the jump to age group was not explained at all. Anyways, in this sense, at a minimum, it should be stated in the manuscript that the findings regarding age group dependency should be changed as “age group-dependent, not age-dependent,” as the authors did not really find the age-dependency per se, but after all the age group dependency as a surrogate.
- In the same vein of reasoning, if the data of pH or [HCO3] are available in the authors’ study, the authors could do similar analyses and show the impreciseness of the relationship using such unreliable data, in order to fortify the correctness of the authors’ assertion utilizing [Cl].
- Lastly, but not least, the rationale or justification of ‘using normalized data’ for further modeling is not given at all, which makes the reading/understanding of such ‘normalized data’ rather difficult to fathom. It is strongly recommended to include rationale(s) for the sake of readers to clearly understand the authors' findings/conclusions.
Author Response
Reviewer 1
The parts changed or added are written in RED in the revised version.
1. Comments by Reviewer 1:
It would be essential to describe as to why the authors chose an independent variable as the age group, rather than age itself. One way to approach this suggestion might be to show the raw data using the age itself as the independent variable, as this reviewer thinks that the jump to age group was not explained at all.
Responses:
The reason why we used age group, rather than age itself: It is possible to present the age-dependent change in [Cl-]s using the age itself. However, it is unclear to see the relationship between the age and [Cl-]s. It is not an unusual, but a usual presentational method to group persons into generations in “clinical studies”. Even if we would account the age of persons at the year (one year) order, it is not accurately to show the age of persons, since the date when persons took standard medical examinations is not just on the age of years old. This means that we have to apply the age group every 1 year old. This application is not essentially different from that shown in the present study. Without grouping in the age, we could not show the mean value at the age. Based on these facts, we would like to present our data with the grouping of persons’ ages every 10 years old in the present study. The result shown in the present study clearly shows that the age is an effective factor influencing [Cl-]s, FBS and HbA1c in the grouping of persons’ age every 10 years old. However, if the editor would suggest us to present our data on ages as continuous numbers, we will do so, although we might be not able to clearly present the age-dependent observation.
Comments by Reviewer 1:
Anyways, in this sense, at a minimum, it should be stated in the manuscript that the findings regarding age group dependency should be changed as “age group-dependent, not age-dependent,” as the authors did not really find the age-dependency per se, but after all the age group dependency as a surrogate.
Responses:
Although we understand the meaning of Reviewer’s suggestion, the usage of ‘age group-dependent’ is unusual, rather not used in clinical studies. Therefore, we would like to use ‘age-dependent’. However, if the editor would suggest us to use ‘age group-dependent’, we will change the term ‘age-dependent’ to ‘age group-dependent’.
2. Comments by Reviewer 1:
In the same vein of reasoning, if the data of pH or [HCO3] are available in the authors’ study, the authors could do similar analyses and show the impreciseness of the relationship using such unreliable data, in order to fortify the correctness of the authors’ assertion utilizing [Cl].
Responses:
If we could have a large number of data of pH or [HCO3-], we are able to show the precise relationship between [Cl-]s and pH / [HCO3-]. Unfortunately, we could not obtain the data of pH or [HCO3-] from a large number of healthy persons used in the present study (n = 107,630) even less numbers such as 10,000, since most of healthy persons take only standard medical examinations.
3. Comments by Reviewer 1:
Lastly, but not least, the rationale or justification of ‘using normalized data’ for further modeling is not given at all, which makes the reading/understanding of such ‘normalized data’ rather difficult to fathom.
Responses:
Please see the comment by Reviewer 2, ‘I suggest that subsections 2.4–2.7 are moved to the separate secretion (Methods) or even Appendix, since this is relatively standard way of computing multiple regression and multiple correlation’. According to the comment by Reviewer 2, we moved subsections 2.4-2.7 to the section of Methods. The normalizing method is not a rare one, but a standard one in the statistical analyzing research area dealing with multiple factors. However, if the editor would suggest us to explain why we used the normalizing method in the present study again, we will do so in detail, although we have already explained this point in text (see lines 127 - 135).
Comments by Reviewer 1:
It is strongly recommended to include rationale(s) for the sake of readers to clearly understand the authors' findings/conclusions.
Responses:
According to Reviewer’s comments, we added some description and Table 8 (page 11) regarding the clinical significance pf the results observed in the present study. Table 8 (page 11) summarizes clinically significant meanings of [Cl-]s.
See lines 202 - 265, and Table 8 (page 11).

Reviewer 2 Report
The manuscript titled »Possibility of Venous Serum Cl- Concentration ([Cl-]s) as a
Marker for Human Metabolic Status: Correlation of [Cl-]s to Age, Fasting Blood Sugar (FBS) and Glycated Haemoglobin (HbA1c)« introduces an interesting idea by which the measured serum Cl- concentration would be a marker of chronic metabolic conditions and would replace technically more challenging measurement of serum bicarbonate concentration, particularly during standard medical examinations. The proposed mechanism involves phenomenon by which the increased serum bicarbonate concentration promotes Cl- output from erythrocytes via anion exchanger. The study also examines correlations of the patient age, fasting glucose concentration and glycated haemoglobin in relation to the plasma [Cl-]. There are some points that should be addressed before further consideration of the manuscript:
- In the introduction (first paragraph), the physiological mechanisms behind the increase in serum Cl- by the anion exchanger are clearly presented. However, in lines 66–77, the authors argue that solely the rate of CO2 production in tissues would affect the plasma HCO3- concentration. This seems unconvincing since the bicarbonate buffer is also regulated by the kidneys and lungs. Doesn't this imply that persons with e.g. high basal metabolism would have increased HCO3- concentration? In this context, the authors should discuss possible causes of (slight) chronic diabetic ketoacidosis, which is acutely normochloremic (high anion gap) acidosis. Also, it is not clear to me, why low CA activity would slow down the reaction in lungs, but not in distal tissues (as stated in lines 76–77).
- Correlations between Cl-/FBS/HbA1c and age are clearly presented but should be further described – particularly the clinical significance of the results (see also point 4). Are the calculated correlations sufficient to conclude that diabetes and mitochondrial dysfunction are indeed the cause for changes in serum [Cl-]?
- I suggest that subsections 2.4–2.7 are moved to the separate secretion (Methods) or even Appendix, since this is relatively standard way of computing multiple regression and multiple correlation. In fact, to improve clarity, I would suggest that the equations are removed where possible. Instead, the procedure/algorithm which was used should be referenced.
- Discussion is unusually long – some of the discussion paragraphs, which are analysing results (e.g. 316– and 337–) should probably be moved to results and rewritten to fit the context. I would also suggest that discussion focuses more on clearly interpreting the results in context with Fig 4.
- In the last part of the discussion (383–430), the authors discuss limitations of the model, but within the context of the proposed theory. Are there any other possible causes of age-dependent [Cl-] increase (e.g. Cl- metabolism in kidneys)?
- Throughout the manuscript, the same mechanism is repeated many times, e.g. in lines 55-65, lines 110-113 and lines 293-305, lines 393-398; this should be shortened where possible.
- Figure 4 should appear earlier, where it is first referenced.
Author Response
Reviewer 2
The parts changed or added are written in RED in the revised version.
1. Comments by Reviewer 2:
In lines 66–77, the authors argue that solely the rate of CO2 production in tissues would affect the plasma HCO3- concentration. This seems unconvincing since the bicarbonate buffer is also regulated by the kidneys and lungs.
Responses:
As the reviewer suggested, we added some description on the roles of the kidneys and lungs in regulation of [HCO3-].
See lines 63 - 65.
Comments by Reviewer 2:
Doesn't this imply that persons with e.g. high basal metabolism would have increased HCO3- concentration? In this context, the authors should discuss possible causes of (slight) chronic diabetic ketoacidosis, which is acutely normochloremic (high anion gap) acidosis.
Responses:
We added some description on ‘possible causes of (slight) chronic diabetic ketoacidosis, which is acutely normochloremic (high anion gap) acidosis’ according to the reviewer’s comment.
Lines 314 - 331.
Comments by Reviewer 2:
Also, it is not clear to me, why low CA activity would slow down the reaction in lungs, but not in distal tissues (as stated in lines 76–77).
Response
According to the comment by Reviewer 2, we changed the description.
See lines 76 - 78, and lines 347 - 361.
2. Comments by Reviewer 2:
Correlations between Cl-/FBS/HbA1c and age are clearly presented but should be further described – particularly the clinical significance of the results (see also point 4).
Responses:
According to the comment by Reviewer 2, we added some description regarding the clinical significance pf the results observed in the present study.
See lines 257 - 265, and Table 8 in page 11.
Comments by Reviewer 2:
Are the calculated correlations sufficient to conclude that diabetes and mitochondrial dysfunction are indeed the cause for changes in serum [Cl-]?
Responses:
It is difficult to definitely conclude that diabetes and mitochondrial dysfunction are indeed the cause for changes in serum [Cl-] from the calculated correlations, since the calculated correlation can generally just suggest the tendency and the possibility as the cause or the result but not lead to any definite conclusion. However, in almost healthy persons without any severe disorders of organs such as the lung, the kidney, etc. on an assumption that the summation of [Cl-]s and [HCO3-]s is constant (almost healthy persons show this phenomenon), we can conclude that diabetes and mitochondrial dysfunction would reflect the value of [Cl-]s in almost healthy persons without severe lung or kidney disorders, since the calculated coefficients had statistical significances as shown in text under the condition that general readers permit us that the results with the statistical significance can confirm the hypothesis.
3. Comments by Reviewer 2:
Subsections 2.4–2.7 are moved to the separate secretion (Methods) or even Appendix, since this is relatively standard way of computing multiple regression and multiple correlation.
Responses:
We moved subsections to ‘Methods’.
See lines 416 - 480.
Comments by Reviewer 2:
The equations are removed where possible. Instead, the procedure/algorithm which was used should be referenced.
Responses:
Reviewer 2 suggested that ‘The equations are removed where possible. Instead, the procedure/algorithm which was used should be referenced.’. However, Reviewer 1 suggested that ‘the rationale or justification of ‘using normalized data’ for further modeling is not given at all, which makes the reading/understanding of such ‘normalized data’ rather difficult to fathom.’. Therefore, we remained equations in the section of Materials and Methods. On the one hand, we removed equations where possible according to the comment by Reviewer 2, we moved sections 2.4 - 2.7 to ‘Materials and Methods’
See lines 416 - 480.
4. Comments by Reviewer 2:
Discussion is unusually long – some of the discussion paragraphs, which are analysing results (e.g. 316– and 337–) should probably be moved to results and rewritten to fit the context. I would also suggest that discussion focuses more on clearly interpreting the results in context with Fig 4.
Responses:
We moved the part of lines 306 - 351 in the previous version of our manuscript to ‘Results’. We also provided description with clear focus and interpretation on the results including the context with Fig. 4. In addition to the description, we added Table 8 (page 11) summarizing clinically significant meanings of [Cl-]s.
See lines 202 - 265, and Table 8 in page 11.
5. Comments by Reviewer 2:
In the last part of the discussion (383–430), the authors discuss limitations of the model, but within the context of the proposed theory. Are there any other possible causes of age-dependent [Cl-] increase (e.g. Cl- metabolism in kidneys)?
Responses:
We added some description regarding aging effects on the kidney function related to Cl- metabolism.
See lines 376 - 382.
6. Comments by Reviewer 2:
Throughout the manuscript, the same mechanism is repeated many times, e.g. in lines 55-65, lines 110-113 and lines 293-305, lines 393-398; this should be shortened where possible.
Responses:
We shorten the description as Reviewer suggested.
We rewrote the part of lines 55 - 65 in the previous version.
See lines 56 - 62.
We rewrote the part of lines 106 - 115 in the previous version.
See lines 106 - 109.
We rewrote the part of lines 293 - 305 in the previous version.
See lines 304 - 313.
We rewrote the part of lines 380 - 399 in the previous version.
See lines 362 - 375.
7. Comments by Reviewer 2:
Figure 4 should appear earlier, where it is first referenced.
Responses:
We moved Figure 4 from page 14 in the previous version to page 10 in the present version.

Round 2
Reviewer 2 Report
I do not have any further comments.